# Patterns of Care for Breast Radiotherapy in Italy: Breast IRRadiATA (Italian Repository of Radiotherapy dATA) Feasibility Study [note 1]

**DOI:** 10.3390/cancers14163927

**Published:** 2022-08-15

**Authors:** Antonella Ciabattoni, Fabiana Gregucci, Giuseppe D’Ermo, Alessandro Dolfi, Francesca Cucciarelli, Isabella Palumbo, Simona Borghesi, Alessandro Gava, Giovanna Maria Cesaro, Antonella Baldissera, Daniela Giammarino, Antonino Daidone, Francesca Maurizi, Marcello Mignogna, Lidia Mazzuoli, Vincenzo Ravo, Sara Falivene, Sara Pedretti, Edy Ippolito, Rosaria Barbarino, Daniela di Cristino, Alba Fiorentino, Cynthia Aristei, Sara Ramella, Rolando Maria D’Angelillo, Icro Meattini, Cinzia Iotti, Vittorio Donato, Silvia Chiara Formenti

**Affiliations:** 1Department of Radiation Oncology, Ospedale San Filippo Neri, ASL Roma 1, 00135 Roma, Italy; 2Department of Radiation Oncology, Ospedale Generale Regionale “F. Miulli”, Acquaviva delle Fonti, 70021 Bari, Italy; 3Department of Surgery, “Pietro Valdoni”, Universitá di Roma “La Sapienza”, 00185 Roma, Italy; 4LILT, Lega Italiana Contro i Tumori, Sede Centrale Via A. Torlonia, 15, 00161 Roma, Italy; 5Department of Radiation Oncology, Azienda Ospedaliera Ospedali Riuniti Marche Nord, 61029 Ancona, Italy; 6Radiation Oncology Section, University of Perugia and Perugia General Hospital, 06123 Perugia, Italy; 7Department of Radiation Oncology, Azienda USL Toscana Sud Est, Sede Operativa Valdarno, 52100 Arezzo Valdarno, Italy; 8Department of Radiation Oncology, Azienda Ospedaliera ULSS 9, 31100 Treviso, Italy; 9Department of Radiation Oncology, Ospedale Bellaria, 40139 Bologna, Italy; 10Department of Radiation Oncology, Azienda Ospedaliera San Camillo Forlanini, 00152 Roma, Italy; 11Department of Radiation Oncology, Centro San Gaetano, Sede di Bagheria e Sede di Mazzara del Vallo, 35121 Palermo e Trapani, Italy; 12Department of Radiation Oncology, Azienda Ospedaliera Ospedali Riuniti Marche Nord, 61029 Pesaro, Italy; 13Department of Radiation Oncology, Azienda Ospedaliera USL Toscana Nord Ovest, 56121 Lucca, Italy; 14Department of Radiation Oncology, Azienda Ospedaliera ASL Viterbo, 01100 Viterbo, Italy; 15Department of Radiation Oncology, Istituto Nazionale Tumori IRCCS Fondazione Pascale, 80131 Napoli, Italy; 16Department of Radiation Oncology, ASST Spedali Civili Brescia, 25123 Brescia, Italy; 17Department of Radiation Oncology, Università Campus Bio-Medico e Fondazione Campus Bio-Medico, 00128 Roma, Italy; 18Department of Radiation Oncology, Fondazione PTV, Policlinico Tor Vergata, 75013 Roma, Italy; 19Department of Experimental and Clinical Biomedical Sciences “M. Serio”, Department of Oncology, Radiation Oncology Unit, Ospedale Universitario Careggi, Universitá di Firenze, 50134 Firenze, Italy; 20Radiation Therapy Unit, Azienda USL—IRCCS di Reggio Emilia, 42122 Reggio Emilia, Italy; 21AIRO President, AIRO-Associazione Italiana di Radioterapia ed Oncologia Clinica, Piazza della Repubblica 32, 20124 Milano, Italy; 22AIRO Past President, AIRO-Associazione Italiana di Radioterapia ed Oncologia Clinica, Piazza della Repubblica 32, 20124 Milano, Italy; 23Department of Radiation Oncology, Weill Cornell Medicine—New York Presbyterian Hospital, New York, NY 10065, USA

**Keywords:** patterns of care, breast cancer, epidemiology, radiotherapy, national register, real-world evidence, clinical practice

## Abstract

**Simple Summary:**

Breast cancer is the most common cancer in women worldwide, with a high prevalence and incidence, configuring an important issue in cancer epidemiology. Over the years, the combination of primary and secondary prevention programs and multidisciplinary treatment approaches has improved the overall survival (OS) and quality of life (QoL) of patients. However, although treatment pathways should be standardized in evidence-based medicine, clinical practice (real-world evidence) may differ from expected. To improve OS and QoL, having a clear picture of the Patterns of Care actually applied is essential. To this aim, Breast IRRADIATA (Italian Repository of RADIotherapy dATA), a collaborative nationwide project, was developed as a simple tool to probe the current pattern of radiotherapy care in Italy and tested in a feasibility study. This pilot feasibility of IRRADIATA encourages a larger application of this tool nationwide and opens the way to the assessment of pattern of care radiotherapy directed to other cancers.

**Abstract:**

*Aim.* Breast IRRADIATA (Italian Repository of RADIotherapy dATA) is a collaborative nationwide project supported by the Italian Society of Radiotherapy and Clinical Oncology (AIRO) and the Italian League Against Cancer (LILT). It focuses on breast cancer (BC) patients treated with radiotherapy (RT) and was developed to create a national registry and define the patterns of care in Italy. A dedicated tool for data collection was created and pilot tested. The results of this feasibility study are reported here. *Methods.* To validate the applicability of a user-friendly data collection tool, a feasibility study involving 17 Italian Radiation Oncology Centers was conducted from July to October 2021, generating a data repository of 335 BC patients treated between January and March 2020, with a minimum follow-up time of 6 months. A snapshot of the clinical presentation, treatment modalities and radiotherapy toxicity in these patients was obtained. A Data Entry Survey and a Satisfaction Questionnaire were also sent to all participants. *Results.* All institutions completed the pilot study. Regarding the Data Entry survey, all questions achieved 100% of responses and no participant reported spending more than 10 min time for either the first data entry or for the updating of follow-up. Results from the Satisfaction Questionnaire revealed that the project was described as excellent by 14 centers (82.3%) and good by 3 (17.7%). *Conclusion.* Current knowledge for the treatment of high-prevalence diseases, such as BC, has evolved toward patient-centered medicine, evidence-based care and real-world evidence (RWE), which means evidence obtained from real-world data (RWD). To this aim, Breast IRRADIATA was developed as a simple tool to probe the current pattern of RT care in Italy. The pilot feasibility of IRRADIATA encourages a larger application of this tool nationwide and opens the way to the assessment of the pattern of care radiotherapy directed to other cancers.

## 1. Introduction

Breast cancer (BC) is the most common cancer in women worldwide [1]. In Italy, the incidence of BC in 2021 was about 55,000 new cases, with an estimated mortality rate of 12,500 events [2]. Over the years, the combination of primary and secondary prevention programs and multidisciplinary treatment approaches has improved both the survival and quality of life of breast cancer patients. New diagnostic, surgical and radiation techniques, more effective systemic treatments, and a better understanding of the disease biology and its complexity have all contributed to this progress [3,4]. The stage of the disease, the patient’s clinical characteristics, the tumor phenotype, and its biology based on biomarker expression are all currently used to inform the individualization of treatment [5]. Radiation oncology has undergone a remarkable technical evolution [6], hand in hand with the surgical revolution that, about 40 years ago, marked the progressive transition from mastectomy toward conservative breast surgical approaches [7]. Radiotherapy (RT) is an important part of BC treatment at all stages of disease, and it has deeply changed over the past twenty years due to modern technologies that allow for more precise treatments, optimal dose distribution and reduction of irradiated volumes of adjacent normal tissue [8,9]. Moreover, a better knowledge of the biology of the tumor, the improvement of the clinical indications and the integration with systemic therapies has allowed the adoption of personalized RT schemes tailored to clinical risk [4,6,7,8,9,10,11]. Importantly, the convergence of technical progress with a better biological insight into the disease has enabled the reduction of the duration of the radiation course in selected patients, from the original 6–7 weeks of daily fractions, Monday to Friday to the current to 3–5 weeks. Another relevant change is the modern management of patient’s follow-up after completion of initial treatment. The care for cancer patients does not end when active treatment is completed [12,13], and particularly for irradiated patients, side effects may manifest later, as medium/long time term sequelae [10]. Heterogeneity of follow-up practices may result in underdiagnosis of late sequelae, as well as delayed detection of recurrence. The last AIRO census [14] reported that approximately 2/3 of the 183 Italian Radiotherapy Centers are located in public general hospitals. Hence, a significant proportion of BC patients receive radiation in non-academic centers. Since the type and quality of breast RT was never assessed before, in the absence of a national register of RT treatments, we defined it as a priority task to remedy this gap. To this end, we developed a tool to permit the assessment of current practices as a basis for informing users about the existing pattern of care and eventually introducing common, standardized guidelines for treatment and follow-up during survivorship [15]. The observation of real-world clinical situations (Real World Evidence, RWE) informs clinicians about the real pattern of care and RWE studies are a complementary partner to Randomized Clinical Trials (RCTs) to generate data on the implementation of high-quality research findings [16] and to generate evidence for future guidelines. The creation of national and international cooperative registers in oncology is aimed at this direction [17,18]. However, since they do not usually include comprehensive RT data, they predictably miss important information [19,20]. We propose the systematic use of a simple, user-friendly data collection tool to study the pattern of RT care in Italy and pilot test this approach to describe the current patterns for breast radiation. The overarching goals are to enhance and standardize the quality of care and its sustainability within the national health system.

Based on this background, a collaborative project supported by the Italian Society of Radiotherapy and Clinical Oncology (AIRO) Breast Group and the Italian League Against Cancer (LILT) has grown and named *Breast IRRADIATA (Italian Repository of RADIotherapy dATA).* The aims of the *Breast IRRADIATA* project are as follows:

(1) To generate an inclusive data collection repository focused on BC patients treated with RT to collect real information from daily clinical practice (RWE);

(2) To assess the prescription and adherence to standard treatments (appropriateness of the therapy prescription and delivery);

(3) To promote the nationwide diffusion of the “*Best Clinical Practice*” in BC RT;

(4) To ensure the best treatment of BC patients, wherever they are treated across the territory.

Here, we report the process for creating the *Breast IRRADIATA* dataset, which began by assessing the feasibility of a pilot study from 17 Italian Radiation Oncology Centers.

## 2. Materials and Methods

### 2.1. Data Collection Creation and Framework

A quick and user-friendly data collection tool was created to allow online data entry and updating in real time. Each participating Radiation Oncologist developed a username and password to log into the system. All data entered was anonymized and its safety and preservation were tested according to European General Data Protection Regulation (GDPR) guidelines [21]. For this study, ethical approval was obtained from the Ethics Committee. From June 2020–June 2021, the informatic system was developed, and since this project was addressed only to Italian Radiation Oncologists, the dataset was written in Italian. 

The *Breast IRRADIATA* template is structured into two parts: the first for the patient’s de-identified demographics (ID) and the insertion of data related to disease diagnosis and treatments; the second for follow-up data. In total, part one includes twenty-one items and part two includes nine items. The items were organized in 6 folders according to the area of interest (Table 1). The graphical architecture of the data collection is shown in Appendix A. A unique alphanumeric code was used to identify each patient inserted into the dataset. Each of the involved clinicians had the ability to view the entered data relating to his/her own patients and export them to an Excel file for any statistical analysis. All other data in the database were also accessible to the participating institutions in a de-identified form. 

### 2.2. Feasibility Study

To validate the applicability of the tool, a pilot-feasibility study to test *Breast IRRADIATA*, promoted within the AIRO Breast Group, was launched in 2021 for a duration of three months, July–October. Data were collected from 17 Italian Radiation Oncology Centers who volunteered to participate in this pilot study. The aims of the pilot study were: (1) to test the feasibility of the system; (2) to evaluate the time needed for entering the data; and (3) to collect suggestions from users to improve *Breast IRRADIATA*. Each participant was asked to retrospectively enter data for 20 consecutive patients treated between January and March 2020 who had at least 6 months of follow-up after completion of RT. In addition, two additional brief questionnaires were sent to all participants: the first one (Data Entry Survey) was sent before starting the data collection, and the second one (Satisfaction Questionnaire) was sent after the data collection was completed. The Data Entry Survey was structured in 9 questions with multiple choice answers (only one answer per question was allowed). The aim of this interview was to evaluate the working patterns of each institution. The Satisfaction Questionnaire was structured in 7 questions: 6 with multiple choice answers (only one answer per question was allowed) and 1 with an open field answer. The aims of the Satisfaction Questionnaire were to evaluate the time spent on compilation, assess the level of satisfaction with the tool and obtain suggestions for improvement. Both questionnaires were accessible online, with a link sent by e-mail to each Italian Radiation Oncology Center participating in the feasibility study. Data and clinical aspects inserted into the dataset and in the two questionnaires were collected and analyzed using descriptive statistics. 

## 3. Results

The 17 Italian Radiation Oncology participating centers were evenly distributed across the national territory, as shown in Appendix B. Eleven centers (65%) were based in general hospitals, 5 (29%) in university hospitals, and 1 (6%) in a scientific institute for research and healthcare.

All institutions completed the pilot study, generating a data repository of a total of 335 cases out of the 340 expected (98.5%). Two centers failed to insert 5 patients, because the COVID-19 pandemics did not allow them to complete the follow-up.

### 3.1. Results of Data Entry and Satisfaction Surveys

Relating to the Data Entry Survey, all questions achieved 100% response. The results are summarized in Table 2.

Regarding the number of breast cancer patients treated per year in each Institution, 3 responding centers (18%) declared that they were treating 50–200 patients per year, 9 centers (53%) 200–500, and 5 centers (29%) more than 500. A multidisciplinary tumor board was routinely conducted in 83% of the centers. As for the RT techniques, the more frequently used technique for whole breast irradiation (WBI) was 3D conformal RT (76%). Where a boost to the tumor bed was indicated, it was administrated as sequential in 64% of the patients and concomitant in 36% of the cases. Partial breast irradiation (PBI) was available in 11 centers (64%) and delivered by external beam RT in 82% of cases. A specific question regarded the use of a Deep Inspiration Breath Hold (DIBH) technique: 2 Institutions (12%) always used DIBH, 10 (59%) used it only in selected cases, and in 5 (29%) never used it. Four of 17 institutions (23.5%) neither used PBI nor DIBH: all of them were General Hospitals. Most participating centers regularly followed their patients after radiation treatment, with intervals of 6–12 months in 59% of the centers and 3–4 months in 12%. In 29% of centers, irradiated patients were seen only if they developed particular needs.

All centers completed the Satisfaction Questionnaire. Eleven centers (64.7%) reported an average time of 2–5 min per patient for entering diagnosis and treatment data and 6 (35.5%) of 5–10 min per patient. A shorter time for entering follow-up data was reported: in 41.2% 1–2 min, in 41.2% 2–5 min, and in 17.6% 5–10 min per patient. No participant reported spending more than 10 min for either the first data entry or for follow-up.

Overall, the project was evaluated in an extremely positive way: 14 centers scored it as excellent (82.3%) and 3 as good (17.7%). In response to suggestions for improving the data collection tool, 6 responders (35.3%) suggested including additional information: specifically, to insert boost data (dose, technique, timing) and inclusion of information about mild degree (G1/G2) as well as the severe toxicities (≥G3). The results of the Satisfaction Survey are summarized in Table 3.

### 3.2. Results of Clinical Data

All patients’ median age of all patients was 64 years (range 30–89 years). Relating to breast surgery, 92% of patients underwent lumpectomy, 7% mastectomy and 1% other approaches. Relating to lymph node surgery, 71% of cases underwent sentinel lymph node biopsy, 21% axillary lymph node dissection and 3% lymph node sampling; in 5% of the cases, the data were missing. Invasive ductal carcinoma was the most common histology (73%), followed by invasive lobular carcinoma (13%), ductal carcinoma in situ (10%) and other (3%); in 1% data were not available. Luminal A subtype characterized 54% of cases, followed by Luminal B HER2 negative (21%), Triple Negative (8%), Luminal B HER2 positive (7%) and HER2 positive (3%). The biological/molecular characterization was not available or not applicable in 3% and 4% of cases, respectively. All centers followed AJCC 2017 (eighth edition) for staging classification: breast cancer was diagnosed as T1 in 68% of cases and N0 in 74%, with a distribution in Stages I, II and III of 56%, 36% and 8%, respectively. No Stage IV cases were reported. Adjuvant RT was performed on 99% of the collected patients. Conventional (50 Gy/25 fractions) and moderately hypofractionated (40.5 Gy/15 fractions or 42.75 Gy/16 fraction) fractionation schedules were used in 40% and 36% of cases, respectively. The RT technique applied in 71% of cases was 3D conformal RT. All patients were treated in the supine position. The median time interval between surgery and adjuvant RT was 13 weeks (range 4–22 weeks). Acute skin toxicity of ≥G3 was detected in 9/335 patients (3%). Regarding systemic therapy, neoadjuvant chemotherapy (CT) was performed in 8% of cases, adjuvant CT in 22%, targeted therapy in 5% and endocrine therapy (ET) in 81%. In this feasibility study, the median follow-up time was 10 months (range 6–15 months) and no local, distant recurrence or death were reported. The clinical characteristics of the study population are summarized in Table 4.

## 4. Discussion

In recent years, several practice-changing studies have been published that significantly impact the recommendation for clinical management of BC [22]. However, to date, one of the greatest challenges of modern oncology is the transferability of “knowledge” and “know-how” from Evidence-Based Medicine (EBM) to RWE [23]. Recommendations derived from the evidence of randomized clinical trials are not always applicable to clinical practice [16,24,25] and their implementation is often delayed. Assessing adherence to these recommendations in RWE is necessary, particularly for common diseases, such as BC. BC management requires a significant commitment to healthcare resources [1,2]. In particular, due to the high clinical and social impact of the therapies, it is necessary to assure access, bringing the state-of-the-art therapies as close as possible to where the patient lives. Due to the high incidence of BC, the collection and systematization of data related to its management is particularly relevant. A coordinated acquisition of healthcare information can make physicians’ work easier and more efficient [23]. In fact, the extensive and systematic storage of information in a correct way permits the analysis of patterns of care and enables interventions to enhance the standardization of decision-making and operative skills. This process results in reduction of possible errors, improved quality of care and economical savings [23]. In the United States, the National Cancer Institute’s SEER (Surveillance, Epidemiology and End Results) collects and publishes data from about 35% of the population [26]; this data collection is publicly accessible. In Italy, the National Tumour Registry collects and updates relevant information regarding the epidemiological and prognostic data of neoplasms [2]. The adoption of data collection tools is increasingly extending to the public health sector. It started with the systematic uptake of programs for diagnostic images and evolved to the adoption of electronic medical records for documenting in- and out-patient medical care, with the aim of improving communication and generating data to inform organizational and financial management strategies [27]. The development of a centralized data repository is particularly interesting in Radiation Oncology, a field that constantly converges clinical, technical and imaging data. It also aims at a universal enhancement of quality of care based on the standardization of procedures and reporting across countries. Some attempts have already been made in this direction, as reported in the literature, such as data collection tools for oncology [28] and radiotherapy [29]. All of them explore the possibility of storing data in Data Warehouses, large repositories, collected by an institution or a structure, managed by a Database Management System (DMS), controlling the organization, storage and retrieval of data for many different and complex needs. Marazzi and colleagues [29] proposed a seven-step Breast LArge DatabasE (BLADE) project, including data collection, analysis and evaluation on a data-sharing platform. It focused on developing a large database of encrypted data available from clinical records and approved by a multidisciplinary team with the aim of standardizing a data collection system. The strength of BLADE lies in its validated variables involving the consensus of a multidisciplinary team of experts; some recognized limits were relatively small and homogeneous sample, which only reflected patients treated at Academic Institutions. The complexity of the systems used could also represent an obstacle to the practical application of this tool in a hospital-based setting. Other wide population-based cancer registries [2,19,26,30,31] or hospital-based registries [32] collect data from cancer patients with the aim of understanding how prevention strategies, treatments and outcomes vary in the population, probing cases that represent the clinical reality of national healthcare. While these tools are highly valuable, they usually fail to adequately capture radiotherapy data.

It is within this conceptual framework that the new software *Breast IRRADIATA* was generated and has shown some promises, as demonstrated by the results from this pilot-feasibility study. As shown in Figure 1, this feasibility study is the starting point of a collaborative network project articulated through several components. The data collected shows consistency with the expected type of invasive BC, which was Luminal A in more than 50%, diagnosed at early stage and usually treated with conservative surgery plus sentinel lymph node biopsy, adjuvant RT and ET, according to stage [33]. Multidisciplinary discussion was performed in 83% of the participating institutions, in line with recommendations for the best breast cancer care [34,35]. Conversely, the latest evidence recommends the standard use of adjuvant hypofractionated RT schemes to breast/chest wall and nodal regions [36]. The results from our pilot assessment showed that 40% of the participants treated their patients with a traditional fractionation scheme (50 Gy/25 fractions). This data is in contrast with the most recent treatment recommendations based on a high level of evidence, and results in unjustified inconvenience to the patient (2 more weeks of daily visits) and in increased and unjustified costs to the health care system. The establishment of national guidelines could overcome this discrepancy. Boost to the tumor bed emerged as another relevant topic in this preliminary national data collection. In 2015, Bartelink and colleagues showed the impact of a boost in improving local control, with the largest absolute benefit in younger patients [37]. Historically, the tumor bed boost has been administered in 5 fractions sequential to the treatment of WBI for a total dose of 10–16 Gy. In recent years, several studies have demonstrated the safety and efficacy of administering the boost during WBI using modern techniques such as Intensity Modulated RT (IMRT) or Volumetric Modulated Arc RT (VMAT) with alternative splitting schemes [38,39,40]. Moreover, boosts should be administrated as intra-operative RT, reporting similar results in terms of efficacy and toxicity compared to external beam techniques [41,42,43,44,45]. The results of the pilot *Breast IRRADIATA* showed that the boost was administrated in only 64% of cases, generally as sequential with a 3D conformal RT technique. The implementation of a concomitant boost could also reduce the number of visits and inconvenience the patient, as well as the cost of treatment. Another point of interest of *Breast IRRADIATA* is the evaluation of toxicity from breast RT. Modern treatment techniques allow good results in local control and survival to be obtained, with limited acute and late toxicity. The analysis of the feasibility study confirmed that only 3% ≥ G3 acute skin toxicity. As suggested by some of the participating centers in our Satisfaction Survey, it is important to expand the toxicity assessment to recording Grades 1–2 in the *Breast IRRADIATA* questionnaire. The overall evaluation of the data collection tool revealed a commitment time for inserting a patient and updating the follow-up that never exceeded 10 min. Our next step is to encourage the participation of the remaining 166 centers to complete Breast IRRADIATA. The results will inform our specialty about our current pattern of care in RT of early breast cancer and guide the development of guidelines for the rapid implementation of available evidence. 

Finally, we believe that the *IRRADIATA* module can easily be adapted to assess patterns of care in other common cancers, such as prostate or lung cancer. These assessments are the first step to help reach optimal care across the territory through standardization of clinical practices.

## 5. Conclusions

Current knowledge for the treatment of high-prevalence diseases, such as BC, has evolved toward a more precise, patient-centered medicine, a rational individualization of care, based on evidence. This sophisticated approach represents a new challenge for oncologists, particularly within the reality of non-academic centers. The first step to assure the adoption of evidence-based medicine is to assess the current situation, beyond the patterns of care at academic institutions, by collecting grassroots information across the territory. To this aim, we have chosen BC RT as a model and have developed a simple data collection tool, *IRRADIATA*, to probe the current pattern of RT care in Italy. Although the sample described in the feasibility study is limited and the application time short, adherence to the data collection and the feedback received from the participants are encouraging in the dissemination of Breast IRRADIATA nationwide.

## Figures and Tables

**Figure 1 cancers-14-03927-f001:**
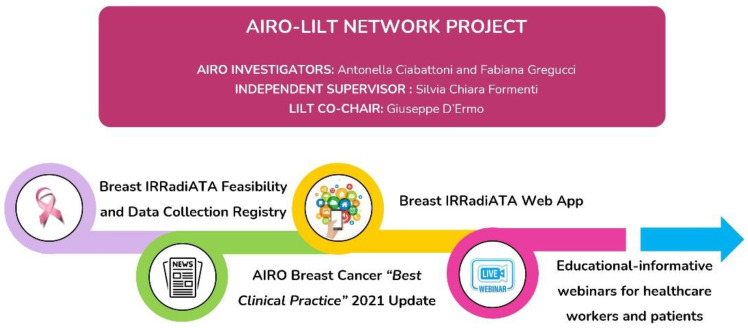
Summary scheme of the *Breast IRRADIATA* project.

**Table 1 cancers-14-03927-t001:** Data collection framework.

Folder	Item	Description	Options
SURGERY	SURGERY_DATA	Specify data of breast surgery	Day/Month/Year
	T_SURGERY_TYPE	Specify type of breast surgery	Lumpectomy
			Mastectomy
			Other
			Not available
	N_SURGERY_TYPE	Specify type of lymph node surgery	Sentinel Lymph node Biopsy
			Axillary Lymph node Dissection
			Other
			Not available
PATHOLOGICAL ANATOMY	cT or pT	Specify the local extent of the disease (clinical or pathological) (AJCC 2017 classification, Eighth edition)	Tx
			Tis
			T1
			T2
			T3
			T4
			Not available
	cN or pN	Specify the lymph node involvement of the disease (clinical or pathological) (AJCC 2017 classification, Eighth edition)	Nx
			N0
			N1
			N2
			N3
			Not available
	HISTOLOGY	Specify the histology of the disease	Ductal Carcinoma In Situ
			Invasive Ductal Carcinoma
			Invasive Lobular Carcinoma
			Other
			Not available
	BIOLOGY	Specify the biology of the disease (St. Gallen 2013 classification)	Luminal A
			Luminal B HER2 negative
			Luminal B HER2 positive
			HER2 positive
			Triple Negative
			Not applicable
			Not available
	GRADING	Specify the grading of the disease	G1
			G2
			G3
			Not available
RADIOTHERAPY	RT_T	Specify whether or not the patient has undergone adjuvant radiotherapy on the breast or chest wall	Yes
			Not
			Not available
	RT_DATA	Specify data of breast radiotherapy	Day/Month/Year
	RT_total_dose	Specify the total dose for whole breast irradiation excluding boost	50 Gy
			40.5 Gy
			42.75 Gy
			30 Gy
			26 Gy
			Other
			Not available
	RT_fractions	Specify the total fractions for whole breast irradiation excluding boost	25
			15
			16
			5
			Other
			Not available
	RT_technique	Specify the technique used for breast irradiation	3D Conformal RadioTherapy
			Intensity Modulated RadioTherapy
			Volumetric Modulated Arc Therapy
			Deep Inspiration Breath Hold
			IntraOperative RadioTherapy
			Partial Breast Irradiation
			Brachytherapy
			Other
			Not available
	RT_LNF	Specify whether or not the patient has undergone adjuvant radiotherapy on the regional lymph node	Yes
			Not
			Not available
ACUTE TOXICITY	A_tox > G3	Specify whether the patient reported acute side effects of a grade equal to or greater than G3 according to the RTOG scale, regardless of the anatomical district, during or at the end of RT	Yes
			Not
			Not available
	A_tox_type	If YES, specify the anatomical district of acute toxicity	Skin
			Soft tissue
			Heart
			Lung
			Lymphedema
			Other
			Not available
SYSTEMIC THERAPY	NACT	Specify whether or not the patient has undergone neoadjuvant chemotherapy	Yes
			Not
			Not available
	ADJUV_CT	Specify whether or not the patient has undergone adjuvant chemotherapy	Yes
			Not
			Not available
	TARGET_tp	Specify whether or not the patient has undergone target therapy	Yes
			Not
			Not available
	HT	Specify whether or not the patient has undergone hormone therapy	Yes
			Not
			Not available
FOLLOW UP	FU_DATA	Specify data of last follow up	Day/Month/Year
	L_tox ≥ G3	Specify whether the patient reported late side effects of a grade equal to or greater than G3 according to the RTOG scale, regardless of the anatomical district, during or at the end of RT	Yes
			Not
			Not available
	L_tox_type	If YES, specify the anatomical district of late toxicity	Skin
			Soft tissue
			Heart
			Lung
			Lymphedema
			Other
			Not available
	Dead	Specify whether the patient is alive or dead	Alive
			Dead
	Dead_DATA	If DEAD, specify data	Day/Month/year
	Loc_relapse	Specify if the patient has developed a local recurrence event (breast, chest wall, regional lymph nodes)	Yes
			Not
			Not available
	Loc_relapse _DATA	If YES, specify data	Day/Month/year
	Distant_mets	Specify if the patient has developed a distant metastases event	Yes
			Not
			Not available
	Distant_mets _DATA	If YES, specify data	Day/Month/year

**Table 2 cancers-14-03927-t002:** Entry survey for a total of 17 Italian Institutes of Radiation Oncology who participated in the study.

Items	Questions	Answers	Responders
Patients per year	In your center, on average, how many breast cancer patients are treated each year?	50–200	3 (18%)
		200–500	9 (53%)
		>500	5 (29%)
Multidisciplinary Board	Are breast cancer patients from your center discussed within a Multidisciplinary Tumor Board?	Yes, ever	8 (48%)
		Yes, in most cases (80–60%)	6 (35%)
		Yes, in minority cases (50–20%)	3 (17%)
		Not, never	0
WBI technique	What is the most commonly used technique for whole breast irradiation to treat these patients?	3DCRT	13 (76%)
		IMRT	3 (18%)
		VMAT	1 (6%)
Boost administration	In your center, if indicated, is the boost delivered sequentially or concomitant?	Sequentially	11 (64%)
		Concomitant (SIB)	3 (18%)
		Both	3 (18%)
PBI	Is partial breast irradiation performed in your center?	Yes	11 (64%)
		Not	6 (36%)
PBI	If yes, with what technique?	EBRT	9 (82%)
		IORT	1 (9%)
		Brachytherapy	1 (9%)
DIBH	In your center, in cases of left breast irradiation, is the Deep Inspiration Breath Hold technique used?	Yes, ever	2 (12%)
		Yes, in selected cases	10 (59%)
		Not, never	5 (29%)
Follow-up	Is the radiotherapy follow-up of patients treated for breast cancer currently performed in your center?	Yes, ever	9 (54%)
		Yes, only in complex cases	4 (23%)
		Yes, only in cases of toxicity	4 (23%)
		Not, never	0
Follow-up	If yes, the follow-up is generally performed how often?	Every 3–4 months	2 (12%)
		Every 6–12 months	10 (59%)
		Variable interval on particular needs	5 (29%)

**Table 3 cancers-14-03927-t003:** Satisfaction Survey for a total of 17 Italian Institutes of Radiation Oncology who participated in the study.

Items	Questions	Answers	Responders
Time-consuming	On average, how long did it take to enter the data relating to the diagnosis and treatment of the individual patient?	1–2 min	0 (0%)
		2–5 min	11 (64.7%)
		5–10 min	6 (35.3%)
		>10 min	0 (0%)
Time-consuming	On average, how long did it take to enter the data relating to the follow-up of the individual patient?	1–2 min	7 (41.2%)
		2–5 min	7 (41.2%)
		5–10 min	3 (17.6%)
		>10 min	0 (0%)
Satisfaction	Report the degree of satisfaction with the operation of the project (ease of access to the site, intuitiveness, and simplicity in filling in the required fields, user-friendly)	Excellent	14 (82.3%)
		Good	3 (17.7%)
		Sufficient	0 (0%)
		Poor	0 (0%)
Satisfaction	Report the degree of satisfaction regarding the relevance of the project (interest in the data collected, completeness of the data collected, relevance of the response options)	Excellent	12 (70.6%)
		Good	5 (29.4%)
		Sufficient	0 (0%)
		Poor	0 (0%)
Satisfaction	Report the degree of satisfaction with the purpose of the project (usefulness in the clinical setting, use for all treated patients, diffusion at national level)	Excellent	13 (76.5%)
		Good	4 (23.5%)
		Sufficient	0 (0%)
		Poor	0 (0%)
Satisfaction	Report the degree of general satisfaction with the project in its entirety	Excellent	14 (82.3%)
		Good	3 (17.7%)
		Sufficient	0 (0%)
		Poor	0 (0%)
Improvement	Are there any aspects of the project that you would improve? If YES, please state which ones and how	Free answer	6 (35.3%) *

* Open responses reported: (1) Add the possibility of specifying boost data (dose, technique, timing) and add the possibility of differentiating the degrees of toxicity. (2) Add the possibility of specifying the timing of systemic therapies. (3) Integrate more clinical information (e.g., type of mastectomy and reconstruction, chemotherapy regimen, patients’ anatomical characteristics) (4) Add the possibility of specifying boost data (dose, technique, timing) (5) Add the possibility of differentiating the degrees of toxicity (6) Enter more useful information about fields filling in the “HELP” function.

**Table 4 cancers-14-03927-t004:** Clinical characteristics of a total of 335 patients added to the data collection.

Folder	Item	Options	Response
SURGERY	T_SURGERY_TYPE	Lumpectomy	309 (92%)
		Mastectomy	25 (7%)
		Other	1 (1%)
		Not available	0
	N_SURGERY_TYPE	Sentinel Lymph node Biopsy	239 (71%)
		Axillary Lymph node Dissection	72 (21%)
		Other	11 (3%)
		Not available	13 (5%)
PATHOLOGICAL ANATOMY	cT or pT	Tx	3 (0.9%)
		Tis	35 (10%)
		T1	229 (68%)
		T2	61 (18%)
		T3	1 (0.2%)
		T4	4 (1%)
		Not available	2 (1.9%)
	cN or pN	Nx	22 (6%)
		N0	248 (74%)
		N1	41 (12%)
		N2	15 (4%)
		N3	6 (3%)
		Not available	3 (1%)
	HISTOLOGY	Ductal Carcinoma In Situ	35 (10%)
		Invasive Ductal Carcinoma	245 (73%)
		Invasive Lobular Carcinoma	45 (13%)
		Other	7 (3%)
		Not available	3 (1%)
	BIOLOGY	Luminal A	180 (54%)
		Luminal B HER2 negative	70 (21%)
		Luminal B HER2 positive	25 (7%)
		Her2 positive	9 (3%)
		Triple Negative	27 (8%)
		Not applicable	13 (4%)
		Not available	11 (3%)
	GRADING	G1	61 (18%)
		G2	168 (50%)
		G3	100 (30%)
		Not available	6 (2%)
RADIOTHERAPY	RT_T	Yes	332 (99%)
		Not	1 (0.3%)
		Not available	2 (0.7%)
	RT_total_dose	50 Gy	136 (40%)
		40.5 Gy	107 (32%)
		42.75 Gy	19 (6%)
		30 Gy	4 (1%)
		26 Gy	10 (3%)
		Other	58 (17%)
		Not available	1 (1%)
	RT_fractions	25	122 (36%)
		15	120 (36%)
		16	32 (10%)
		5	18 (5%)
		Other	40 (12%)
		Not available	3 (1%)
	RT_technique	3D Conformal RadioTherapy	239 (71%)
		Intensity Modulated RadioTherapy	19 (6%)
		Volumetric Modulated Arc Therapy	35 (10%)
		Deep Inspiration Breath Hold	4 (2%)
		IntraOperative RadioTherapy	0
		Partial Breast Irradiation	14 (4%)
		Brachytherapy	0
		Other	24 (7%)
		Not available	0
	RT_LNF	Yes	36 (10%)
		Not	298 (89%)
		Not available	1 (1%)
ACUTE TOXICITY	A_tox > G3	Yes	9 (3%)
		Not	312 (93%)
		Not available	14 (4%)
	A_tox_type	Skin	7 (78%)
		Soft tissue	1 (11%)
		Heart	0
		Lung	0
		Lymphedema	1 (11%)
		Other	0
		Not available	0
SYSTEMIC THERAPY	NACT	Yes	25 (8%)
		Not	295 (88%)
		Not available	15 (4%)
	ADJUV_CT	Yes	72 (22%)
		Not	245 (73%)
		Not available	18 (5%)
	TARGET_tp	Yes	18 (5%)
		Not	294 (88%)
		Not available	23 (7%)
	HT	Yes	268 (81%)
		Not	52 (15%)
		Not available	15 (4%)
FOLLOW UP	L_tox ≥ G3	Yes	0
		Not	327 (98%)
		Not available	8 (2%)
	L_tox_type	Skin	0
		Soft tissue	0
		Heart	0
		Lung	0
		Lymphedema	0
		Other	0
		Not available	0
	Dead	Alive	335 (100%)
		Dead	0
	Loc_relapse	Yes	0
		Not	335 (100%)
		Not available	0
	Distant_mets	Yes	0
		Not	335 (100%)
		Not available	0

## Data Availability

Research data are stored in an institutional repository and will be shared upon request to the corresponding author.

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
