# Peer review of "Patterns of Care for Breast Radiotherapy in Italy: Breast IRRadiATA (Italian Repository of Radiotherapy dATA) Feasibility Studyâ€"

_cancers, 2022, doi:10.3390/cancers14163927_

Round 1
Reviewer 1 Report
Data collection and its correct evaluation a understanding is most important to fully using of all possibility of modern treatment. The main deal if this research, actually it is not research at all, is new system data collection feasibility study and its user friendliness. In mi point of view the modern trends in radiotherapy produce many data and it is very hard to mine some valuable information from this data.
There is the lack of outcomes from this system, which could the authors consider regarding the methodology.The conclusions consistent with the evidence and arguments presented and they address the main question posed.It is a tool how to help implement the evidence based medicine into the practice.
Author Response
Thank you very much for your manuscript revision and critical help to improve it. We agree with the reviewer's comments. Data collection aimed at improving care pathways is an important goal of modern oncology, in particular the observation of real-world clinical situations (Real World Evidence, RWE) informs physicians about the real model of care and RWE studies are a complementary partner to Randomized Clinical Trials (RCTs) to generate data on the implementation of high-quality research results (as highlighted in the text).
This manuscript presents the results of a feasibility study relating to the running in of a system developed for simple data collection that can be implemented in all radiotherapy centers.
As reported in methodology: “the aims of the pilot study were: 1) to test the feasibility of the system; 2) to evaluate the time needed for entering the data; 3) to collect suggestions from users in order to improve IRRADIATED Breast.”.
In Results, we reported that "Overall, the project was valuated in an extremely positive way: 14 centers scored it as excellent (82.3%) and 3 as good (17.7%). Eleven centers (64.7%) reported an average time of 2 -5 minutes per patient for entering diagnosis and treatment data and 6 (35.5%) of 5-10 minutes per patient. A shorter time for entering follow-up data was reported: in 41.2% 1-2 minutes, in 41.2% 2- 5 minutes and 17.6% 5-10 minutes per patient. No participant reported spending more than 10 minutes for either the first data entry or for follow up. In response to suggestions for improving the data collection tool, 6 responders (35.3%) suggested to include additional information: specifically, to insert boost data (dose, technique, timing) and inclusion of information about mild degree (G1 / G2) as well as the severe toxicities (≥G3). Results of the Satisfaction Survey are summarized in Table 3 . ".
Systematic data collection is being implemented, which will be as representative as possible of the real ongoing care pathways.
Reviewer 2 Report
In the manuscript titled “Patterns of Care for Breast Radiotherapy in Italy: Breast IRRadiATA* (Italian Repository of Radiotherapy dATA) Feasibility Study”, the author has established a simple data collection tool (IRRADIATA) to probe the current pattern of breast cancer RT care in Italy. This pilot study evaluated the feasibility of IRRADIATA in 17 Italian Radiation Oncology Centers and showed promising results. Overall, the paper is worthy of publication in Cancers, while due to the small number of participants, it still needs to be evaluated over a longer period of time to verify the universal applicability and value of the IRRADIATA. In addition, the English grammar, punctuation, and spelling need to be carefully corrected.
Author Response
Thank you very much for your manuscript revision and critical help to improve it. We agree that although the present research shows positive results a greater number of participants and a longer application period of time are necessary to evaluate the applicability. This aspect was highlighted in Conclusions: “Although the sample described in the feasibility study is limited and the application time short, the adherence to the data collection and the feedback received from the participants are encouraging in the dissemination of Breast IRRADIATA nationwide”.
English revision has been made.
Reviewer 3 Report
Dear authors,
I read your paper with a great interest and find it acceptable at it stands. The main question addressed by the research is providing an online database to detect how breast cancer patients in Italy have been receiving adjuvant radiotherapy. Despite the validity of hypofractionated radiotherapy schedule, these treatments are underestimated due to either financial aspect or uncertainties regarding the adverse events and efficacy. Using such databases will help to shed light on various aspects of adjuvant radiotherapy in the management of breast cancer.
I suggest to:
- consider the settings where patients are treated (private section, public sector, academic centers)
- consider the treatment volumes
- consider the actual treatment time
- consider the financial aspect (direct and indirect)
Some suggestions to be cited in the discussion to improve the quality of it.
1.Fadavi P, Nafissi N, Mahdavi SR, Jafarnejadi B, Javadinia SA. Outcome of hypofractionated breast irradiation and intraoperative electron boost in early breast cancer: A randomized non-inferiority clinical trial. Cancer Rep (Hoboken). 2021 Oct;4(5):e1376. doi: 10.1002/cnr2.1376. Epub 2021 Apr 1. PMID: 33797199; PMCID: PMC8552001.
2. Homaei Shandiz, F., Fanipakdel, A., Forghani, M.N. et al. Clinical Efficacy and Side Effects of IORT as Tumor Bed Boost During Breast-Conserving Surgery in Breast Cancer Patients Following Neoadjuvant Chemotherapy. Indian J Gynecol Oncolog 18, 46 (2020). https://doi.org/10.1007/s40944-020-00389-5
3. Keramati, A., Javadinia, S.A., Gholamhosseinian, H. et al. A Review of Intraoperative Radiotherapy After Neoadjuvant Chemotherapy in Patients with Locally Advanced Breast Cancer: From Bench to Bedside. Indian J Gynecol Oncolog 18, 110 (2020). https://doi.org/10.1007/s40944-020-00465-w
Please be informed that these papers have been suggested to improve the manuscript quality in accordance with the most up-to-date literature and there was no intention to induce citation and you can use them optionally based on your preference.
Author Response
Thank you very much for your manuscript revision and critical help to improve it. Your suggestion regarding the possibility of evaluating the impact of the financial aspect on the implementation of alternative hypofractionated radiotherapy schedule, as well as the uncertainties relating to side effects, is certainly a fundamental point to consider. This aspect is being verified in a subsequent study of wide involvement on the national territory.
The suggested references have been added to the text.